# Amphiphilic Poly(*N*-vinylpyrrolidone) Nanoparticles Conjugated with DR5-Specific Antitumor Cytokine DR5-B for Targeted Delivery to Cancer Cells

**DOI:** 10.3390/pharmaceutics13091413

**Published:** 2021-09-07

**Authors:** Anne Yagolovich, Andrey Kuskov, Pavel Kulikov, Leily Kurbanova, Dmitry Bagrov, Artem Artykov, Marine Gasparian, Svetlana Sizova, Vladimir Oleinikov, Anastasia Gileva, Mikhail Kirpichnikov, Dmitry Dolgikh, Elena Markvicheva

**Affiliations:** 1Shemyakin-Ovchinnikov Institute of Bioorganic Chemistry of the Russian Academy of Sciences, 117997 Moscow, Russia; leyli.kurbanova_1997@mail.ru (L.K.); art.al.artykov@gmail.com (A.A.); marine_gasparian@yahoo.com (M.G.); sv.sizova@gmail.com (S.S.); voleinik@mail.ru (V.O.); sumina.anastasia@mail.ru (A.G.); lemarkv@hotmail.com (E.M.); kirpichnikov@inbox.ru (M.K.); dolgikh@nmr.ru (D.D.); 2Faculty of Biology, Lomonosov Moscow State University, 119234 Moscow, Russia; bagrov@mail.bio.msu.ru; 3D. Mendeleev University of Chemical Technology of Russia, 125047 Moscow, Russia; ankuskov@mail.ru; 4Federal State Budgetary Institution “Centre for Strategic Planning and Management of Biomedical Health Risks” of the Federal Medical Biological Agency, 119121 Moscow, Russia; p.kulikov.p@gmail.com

**Keywords:** poly(*N*-vinylpyrrolidone), self-assembly, nanoparticles, targeted drug delivery system, DR5 receptor, TRAIL variant DR5-B, multicellular tumor spheroids, antitumor therapy

## Abstract

Nanoparticles based on the biocompatible amphiphilic poly(*N*-vinylpyrrolidone) (Amph-PVP) derivatives are promising for drug delivery. Amph-PVPs self-aggregate in aqueous solutions with the formation of micellar nanoscaled structures. Amph-PVP nanoparticles are able to immobilize therapeutic molecules under mild conditions. As is well known, many efforts have been made to exploit the DR5-dependent apoptosis induction for cancer treatment. The aim of the study was to fabricate Amph-PVP-based nanoparticles covalently conjugated with antitumor DR5-specific TRAIL (Tumor necrosis factor-related apoptosis-inducing ligand) variant DR5-B and to evaluate their in vitro cytotoxicity in 3D tumor spheroids. The Amph-PVP nanoparticles were obtained from a 1:1 mixture of unmodified and maleimide-modified polymeric chains, while DR5-B protein was modified by cysteine residue at the N-end for covalent conjugation with Amph-PVP. The nanoparticles were found to enhance cytotoxicity effects compared to those of free DR5-B in both 2D (monolayer culture) and 3D (tumor spheroids) in vitro models. The cytotoxicity of the nanoparticles was investigated in human cell lines, namely breast adenocarcinoma MCF-7 and colorectal carcinomas HCT116 and HT29. Notably, DR5-B conjugation with Amph-PVP nanoparticles sensitized resistant multicellular tumor spheroids from MCF-7 and HT29 cells. Taking into account the nanoparticles loading ability with a wide range of low-molecular-weight antitumor chemotherapeutics into hydrophobic core and feasibility of conjugation with hydrophilic therapeutic molecules by click chemistry, we suggest further development to obtain a versatile system for targeted drug delivery into tumor cells.

## 1. Introduction

Presently, nanocarriers are increasingly developed for the treatment of solid tumors [1]. Polymeric micellar nanoparticles are of great interest due to their high stability, tunable properties and the ability for multi-drug delivery [2,3,4]. A number or polymeric micelles with active tumor-targeting properties were proposed for drug delivery, for example, c(RGDyK)-linked poly(carboxybetaine methacrylate)-poly(crcaprolactone)-based [5], cRGD-conjugated poly(ethylene glycol)-*b*-poly(l-glutamic acid)-based [6], citric acid-functionalized poly(ε-caprolactone)-based micelles [7]. Poly(*N*-vinylpyrrolidone) is non-toxic, non-carcinogenic, eco-friendly, FDA-approved biocompatible water-soluble polymer with a rather long history of pharmaceutical use [8]. Amphiphilic poly(*N*-vinylpyrrolidone) derivatives (Amph-PVP) self-aggregate in aqueous solutions with the formation of micellar nanoscaled structures, which are capable to entrap various therapeutic molecules under mild conditions [9]. The Amph-PVP nanoparticles show serum stability and excellent blood compatibility [10]. Amph-PVP tunable properties include the control of nanoparticle structure, size and morphology combined with the simple, low-cost, scalable production, making them suitable for a variety of biomedical applications [11,12,13], primarily drug delivery [14,15,16].

TRAIL is promising for targeted anticancer therapy due to its tumor selectivity and high safety profile [17]. Despite disappointing results in the first set of clinical trials, TRAIL finally passed phase III with vinorelbine and cisplatin in patients with advanced non-small-cell lung cancer, showing moderate antitumor activity [18]. TRAIL-based antitumor therapy holds promise for clinical implementation due to its tumor selectivity and safety. However, a short half-life and thus the insufficient apoptosis-inducing capacity obstruct TRAIL use in the clinic [19]. Therefore, various strategies were proposed to enhance the TRAIL stability and antitumor activity [20].

Recently, we developed TRAIL DR5 receptor-specific variant DR5-B, which has no affinity neither to DR4 nor to decoy receptors DcR1, DcR2, and osteoprotegerin and with enhanced pro-apoptotic activity in tumor cells [21]. DR5-B was shown to inhibit the growth of HCT116 colorectal cancer xenografts more efficiently compared to TRAIL and to increase animal survival [22]. The main advantage of DR5-B over the wild-type TRAIL consists in overcoming the receptor-dependent TRAIL-resistance of tumor cells. However, the pharmacokinetic parameters of DR5-B were similar to those of TRAIL, except that the DR5-B half-life was 3.5-fold higher than that one of TRAIL due to the increased selectivity to DR5 leading to lower tissue binding [22]. Thus, despite the increased DR5-B antitumor efficiency due to the DR5 specificity, its fast elimination from the body may hinder implementation to clinical practice In spite of the development of a number of nano delivery systems for wild-type TRAIL [23], DR5-specific antitumor cytokine DR5-B has not ever been conjugated to any type of nanocarriers.

The aim of the current study was to prepare Amph-PVP-based nanoparticles covalently conjugated with DR5-specific antitumor cytokine, namely TRAIL mutant variant DR5-B, and to evaluate their in vitro cytotoxicity in 3D tumor spheroids. Here, we obtained new Amph-PVP-based self-assembled nanoparticles covalently conjugated with the DR5-specific antitumor cytokine DR5-B. The cytotoxicity of the nanoparticles was studied in 2D and 3D in vitro models using human breast adenocarcinoma MCF-7 cells, human colorectal carcinoma HCT116 and colorectal adenocarcinoma HT29 cells. To our knowledge, this is the first attempt to obtain Amph-PVP-based nanoparticles coated with the covalently linked targeted protein molecules.

## 2. Materials and Methods

### 2.1. Materials, Reagents and Cell Lines

*N*-vinyl-2-pyrrolidone (VP), 2,2’-azobisisobutyronitrile (AIBN), 1,4-dioxane were obtained from Acros (Geel, Belgium). Prothionamide, stearoyl chloride, maleimide, potassium tert-butylate, dimethylsulfoxide (DMSO), fetal bovine serum (FBS), phosphate buffer saline (PBS), and all other chemicals were obtained from Sigma-Aldrich (Saint Louis, MO, USA) unless otherwise specified. All solvents and components of buffer solutions were of analytical grade and used as received. Milli-Q Plus System (Merck KGaA, Darmstadt, Germany) was used for distilled–deionized water preparation.

Human colorectal carcinoma HCT116 cells, breast adenocarcinoma MCF-7 cells, normal keratinocytes HaCaT and skin fibroblasts BJ-5ta were from ATCC (Washington, WA, USA); human colorectal adenocarcinoma HT-29 was kindly provided by Institute of Cytology, Russian Academy of Sciences (St. Petersburg, Russia). Cell culture media (DMEM, RPMI1640), 0.05% trypsin-EDTA solution, and phosphate-buffered saline tablets were from PanEco (Moscow, Russia). Fetal bovine serum was from HyClone (Cytiva, Marlborough, MA, USA). MTT reagent was from Sigma-Aldrich (Saint Louis, MO, USA). Cyclic RGD-peptide, namely cyclo-RGDfK(TPP) for generation of multicellular tumor spheroids was kindly provided by Prof. Burov (St-Petersburg, Russia).

### 2.2. Expression and Purification of Recombinant Protein DR5-B

For covalent linking with Amph-PVP nanocarriers, recombinant protein DR5-B was genetically modified by replacing amino acid residue valine in 114 position to cysteine (DR5-B/V114C) by site-directed mutagenesis. The oligonucleotide primers were *ggaaggagatatacatatgtgcagagaaagaggtcctc* and *gaggacctctttctctgcacatatgtatatctccttcc*. Introduction of mutations was confirmed by DNA sequencing. The highly purified recombinant protein DR5-B/V114C was obtained from *E. coli* SHuffle B strain as described previously [24]. Briefly, the cells were transformed by plasmid pET32a/sdr5-*b*-v114c. Cell culture was grown at 28 °C for 20 h. Cells were disrupted by French Press (Spectronic Instruments Inc., Texas City, TX, USA) under a pressure of 2000 psi. DR5-B/V114C protein was purified from the soluble fraction of cytoplasmic proteins by immobilized metal affinity chromatography on Ni-NTA agarose (Qiagen, Hilden, Germany), followed by ion-exchange chromatography on SP Sepharose (GE Healthcare, Chicago, IL, USA). Protein preparation was dialyzed against 50 mM NaH_2_PO_4_, 150 mM NaCl, pH 7.0 for 24 h at 4 °C, sterilized by filtration, lyophilized, and stored at −70 °C.

### 2.3. Production of P-DR5-B Nanoparticles

#### 2.3.1. Synthesis of Amph-PVP-Cl (OD11000)

The amphiphilic PVP containing one end hydrophobic stearoyl group was prepared by radical polymerization of monomer in dioxane with azobisisobutyronitrile as an initiator and stearoyl chloride as a chain-transfer agent and chain-growth regulator, as described earlier [25]. Estimated quantities of monomer, initiator and stearoyl chloride were dissolved in dioxane and polymerization reaction was carried out for 2 h at 80 °C.

Next, the obtained polymer was mixed with a five-fold volume of ultrapure water. The prepared polymer solution was purified by dialysis against ultrapure water using Slide-A-Lyzer Dialysis Cassettes (Thermo Fisher Scientific, Waltham, MA, USA) with a molecular weight cut-off (MWCO) of 12,000 Da. The medium conductivity (µS/cm) was measured every 6 and 24 h, and dialysis water was checked for impurities spectrophotometrically and by HPLC. The water was changed every 24 h. After 3 days no significant reduction of the conductivity and no presence of any impurities were observed, and the dialysis was terminated. The purified polymer solutions were subsequently freeze-dried (Martin Christ, Osterode am Harz, Germany) and used for further experiments. The yield of the polymers was about 78–89%. The number-average molecular weight (Mn) of the amphiphilic PVP was determined by functional analysis (titration) and, alternatively, in toluene solutions using vapor pressure osmometer (Knauer, Berlin, Germany) and polystyrene standards. In the current study, the obtained amphiphilic polymer with molecular weight of 11,000 Da (PVP-OD11000) was used.

#### 2.3.2. Potassium Maleimide Synthesis

For the synthesis of potassium maleimide, a mixture of maleimide and potassium tert-butylate (with a molar excess of 1.2) was stirred at 0 °C for 1 day in the dimethoxyethane medium, followed by precipitation of potassium maleimide into dimethyl ether.

#### 2.3.3. Modification of Amph-PVP-Cl by Maleimide Group

A maleimide group was attached to the synthesized homopolymer by reacting an amphiphilic *N*-vinyl-2-pyrrolidone homopolymer containing chlorine at the end of the hydrophilic chain (Amph-PVP-Cl) with potassium maleimide. The polymer with a maleimide active group was synthesized in the dioxane medium as follows: dry dioxane, the Amph-PVP-Cl polymer, and potassium maleimide (with a molar excess of 1.1) were stirred at room temperature for 1 h, diluted with distilled water, dialyzed against water for 3 days and lyophilized.

#### 2.3.4. Amph-PVP Nanoparticles Preparation

For nanoparticle preparation, various molar ratios of modified Amph-PVP-Maleimide and non-modified Amph-PVP-Cl (PVP-OD11000) polymers (1:3 and 1:1) were used. Micelles with model substance prothionamide in the hydrophobic core were prepared by an emulsification method. For this purpose, 0.4 g of the polymer mixture was dispersed in 50 mL of water and 0.005 g of prothionamide was dissolved in 2 mL of chloroform. Both solutions were mixed under ultrasonic treatment (undercooling, 12 min). After homogenization, chloroform was distilled off on a rotary evaporator Heidilph Hei-Vap Value Digital (Schwabach, Germany). To separate prothionamide which was not included in the nanoparticles, the suspension was centrifuged (4000 g, 5 min) using Sigma 4–5L (Osterode am Harz, Germany). Then the supernatant was frozen and lyophilized.

#### 2.3.5. Conjugation of the Amph-PVP Nanoparticles with DR5-B

The DR5-B/V114C protein was conjugated with Amph-PVP nanoparticles by selective covalent interaction of the cysteine residue 114 of the protein with the Amph-PVP-Maleimide included in the nanoparticles. Conjugation was carried out in PBS (pH 7.3) at room temperature overnight, followed by centrifugation for 10 min at 10,000× *g* and resuspension of the precipitates in PBS (pH 7.3). To determine the sorption capacity, the particles were diluted in PBS (pH 7.3) at concentration of 5 mg/mL and incubated at room temperature overnight with various amounts of the DR5-B/V114C protein. The mixtures were centrifuged for 10 min at 10,000× *g* and the protein levels in the supernatants as well as in resuspended sediments were determined by Bradford assay.

The particle size distribution and polydispersity index (PDI) were determined by dynamic light scattering (DLS) on Brookhaven Instruments Corporation 90Plus Particle Analyzer (Holtsville, NY, USA) with 90Plus Particle Sizing Software (version 5.41, Brookhaven Instruments Corporation, Holtsville, NY, USA).

### 2.4. Imaging Nanoparticles by TEM and AFM

The Amph-PVP nanoparticles were imaged using transmission electron microscopy (TEM) and atomic force microscopy (AFM). For the TEM imaging, carbon-coated grids (Ted Pella, Redding, CA, USA) were treated using a glow discharge device Emitech K100X (Quorum Technologies Ltd., Lewes, UK) at 25 mA. The suspension of the nanoparticles was deposited onto the grid for 0.5–1 min, and the grids were stained with 1% uranyl acetate for 1–2 min. Images were obtained using a JEM-1011 (Jeol, Tokyo, Japan) transmission electron microscope equipped with an Orius SC1000W camera (Gatan Inc., Pleasanton, CA, USA). The acceleration voltage was at 80 kV; image processing was carried out using Fiji software [26].

For the AFM imaging, the samples were deposited onto mica sheets coated with amorphous carbon; coating was carried out using a Hitachi HUS-5GB device (Hitachi, Tokyo, Japan). As for the TEM imaging, the substrate surface was treated with a glow discharge device before sample deposition. The samples were deposited for 0.5–1 min and quickly dried with a stream of nitrogen. Imaging was carried out using a Solver PRO-M microscope (NT-MDT, Zelenograd, Russia). The images were acquired in semicontact mode with NSG10 cantilevers (nominal tip curvature radius 6 nm) at 1–1.4 Hz. Image processing was carried out using Gwyddion (version 2.53, http://gwyddion.net/ (accessed on 7 September 2021)) [27].

### 2.5. Cell Culture and Multicellular Tumor Spheroids Formation

Human colorectal carcinoma HCT116 cells, human breast adenocarcinoma MCF-7 cells, human keratinocytes HaCaT and human skin fibroblasts BJ-5ta were cultured in DMEM supplemented with 10% FBS in a 5% CO_2_ humidified atmosphere at 37 °C. Colorectal adenocarcinoma HT-29 cells were cultured in RPMI-1640 supplemented with 10% FBS at 37 °C and 5% CO_2_. The cells were detached after treatment with trypsin-EDTA solution (0.25% *v*/*v*), and the culture medium was replaced every 3–4 days. Multicellular tumor spheroids were produced by the RGD-induced cell self-assembly method, as described earlier [28]. Briefly, cells (50,000 cells/mL) were seeded in a 96-well plate (100 μL/well) and incubated at 37 °C for 2–3 h until the cells attached to the plate bottom. Then the medium was replaced in each well with 100 μL of complete DMEM (10% FBS) containing cyclo-RGDfK(TPP) peptide (40 μM). Finally, the cells were transferred to a CO_2_ incubator, and RGD-induced spheroid formation was observed in 72 h.

### 2.6. Cytotoxicity Evaluation

Cytotoxicity was evaluated by MTT-assay. The tumor spheroids were generated with cyclo-RGDfK(TPP) as described in Section 2.5. Preparations at various dilutions were added to each well, and the cells were transferred to the CO_2_-incubator for 24 or 48 h. After treatment, the cells were stained with a 0.05% (*w*/*v*) MTT solution in DMEM for 4 h. Then medium was replaced with DMSO (100 μL/well) and absorbance (570 nm) was measured by Multiskan FC reader (Thermo Fisher Scientific, Waltham, MA, USA). The cell viability was determined in % compared to the control according to the equation: (OD sample—OD background)/(OD control—OD background) × 100%. The half-maximal inhibitory concentration (IC50) was determined as drug concentration resulting in 50% inhibition of cell growth.

### 2.7. Statistical Analysis

Cell culture experiments were repeated at least three times. The data were normally distributed and displayed as mean ± SD from at least three replicates. GraphPad Prism version 6.01 (GraphPad Software Inc, San Diego, CA, USA) for Windows was used to generate graphical representations and conduct statistical analyses of data. Comparisons between two groups from 2–3 independent experiments were made by one-way analysis of variance at 95% confidence using the unpaired *t*-test and Bonferroni’s multiple comparison tests or uncorrected Fisher’s least significant difference for comparisons between more than two groups. IC50 values were calculated in GraphPad Prism according to the built-in dose-response inhibition formula.

## 3. Results

### 3.1. Preparation of Amph-PVP Nanoparticles

Schemes of the synthesis of compounds for the preparation of the Amph-PVP nanoparticles are shown in Figure 1. The amphiphilic *N*-vinylpyrrolidone polymers were synthesized by the originally developed one-step method reported by us earlier [25,29]. Specifically, the amphiphilic PVP containing one end stearoyl group (Amph-PVP-Cl) was prepared by radical polymerization of monomer in dioxane with azobisisobutyronitrile as an initiator and stearoyl chloride as a chain-transfer agent and chain-growth regulator) (Figure 1a). Additionally, Amph-PVP-Cl molecules were modified with maleimide at the hydrophilic ends of the polymer chains for further covalent conjugation with DR5-B. For this purpose, potassium maleimide was synthesized (Figure 1b) and a maleimide group was attached to Amph-PVP-Cl by reacting with amphiphilic *N*-vinyl-2-pyrrolidone homopolymer containing chlorine at the end of the hydrophilic chain (Amph-PVP-Cl) (Figure 1c). The core of the self-assemblies was loaded with hydrophobic model substance prothionamide, an anti-tuberculosis drug without antitumor activity. Nanoparticles were obtained by the emulsion method by joint ultrasonication of prothionamide solution in chloroform and Amph-PVP solution in water.

We obtained two variants of self-assembled nanoparticles from mixtures of maleimide-modified and unmodified Amph-PVPs with molar ratios 1:3 and 1:1 for further optimization of the amount of the bound protein.

### 3.2. Conjugation of the Amph-PVP Nanoparticles with the DR5-B Protein

The DR5-B protein is the DR5 receptor-specific mutant variant of the soluble extracellular domain of TRAIL (114-281) cytokine with six amino acid substitutions. For covalent conjugation with maleimide groups of the Amph-PVP nanoparticles, the DR5-B mutant variant with N-terminal V114C substitution, namely DR5-B/V114C, was obtained by the site-specific mutagenesis. DR5-B/V114C protein in *E. coli* strain SHuffle B and purified by the recently developed method for high-throughput production of recombinant TRAIL variants, as described previously [24] (see Appendix A). The DR5-B/V114C cytotoxicity in human colorectal carcinoma HCT116 cells was similar to this one of DR5-B (Appendix A) which confirmed that N-terminal V114C substitution did not interfere with the DR5-B cytotoxic activity.

DR5-B/V114C was conjugated to the surface of the Amph-PVP nanoparticles by the selective covalent interaction of cysteine residue 114 with maleimide-modified Amph-PVP, as described in the Materials and Methods section. As a result, the DR5-*B*-conjugated Amph-PVP nanoparticles (further called P-DR5-B) were fabricated. Sorption capacity amounted up to 5 μg of the DR5-B protein per 1 mg of the lyophilized nanoparticles (Table 1). Expectedly, the protein sorption capacity slightly increased with an increase in maleimide-modified Amph-PVP contents. However, this dependence was non-linear, which is obviously associated with steric effects upon protein binding to the nanoparticle surface. Contents of maleimide-modified Amph-PVP and the protein attachment did not influence the nanoparticle median hydrodynamic size, which was 220 nm, as measured by DLS (Table 1, Figure 2d). Polydispersity index evidenced that both the Amph-PVP and P-DR5-B samples were unimodal (Table 1) since in both cases PDI was equal to or lower than 0.3 [30,31]. A further study was carried out with the P-DR5-B sample containing 1:1 maleimide-modified to unmodified Amph-PVP contents due to its optimal DR5-B protein sorption capacity.

The P-DR5-B nanoparticles were visualized by atomic force microscopy and transmission electron microscopy (Figure 2a,b, respectively). Using both imaging techniques, we observed the round-shaped nanoparticles. We used carbon-coated mica as a substrate for the AFM imaging because upon the glow discharge treatment, it provided better sample adsorption than the other commonly used substrates (e.g., mica or Ni+2-treated mica). The low height of the nanoparticles (typically below 15 nm) indicated severe flattening, as commonly observed with AFM. The diameters of individual nanoparticles measured by AFM and TEM were in good agreement, namely, the means were D_AFM_ = 140 ± 90 nm and D_TEM_ = 110 ± 50 nm (Figure 2c) and in a reasonable agreement with the DLS data. The median hydrodynamic size of the P-DR5-B nanoparticles measured by DLS was about 220 nm (Table 1, Figure 2d). The difference between the sizes calculated using microscopy and DLS could arise due to the sample polydispersity and the relatively low sensitivity of DLS towards the small nanoparticles.

### 3.3. Cytotoxicity Study of the P-DR5-B Nanoparticles in 2D and 3D In Vitro Models

The in vitro cytotoxicity of the P-DR5-B nanoparticles was evaluated in 2D (monolayer culture) and 3D (tumor spheroids) models. For this purpose, human breast adenocarcinoma MCF-7 cells, human colorectal carcinomas HCT116 cells and human colorectal adenocarcinoma HT29 cells were used. The tumor spheroids were generated by a simple RGD-induced cell self-assembly technique recently developed by the authors [28]. Expectedly, MCF-7 cells in the 2D model (Figure 3a) were more sensitive to the P-DR5-B nanoparticles compared to those in the spheroids (Figure 3b, Table 2).

Therefore, the further cytotoxicity study of the P-DR5-B nanoparticles for HCT116 and HT29 cells were carried out only in the 3D model (Figure 4), since the spheroids are considered as a more relevant model than monolayer culture due to better mimicking small solid tumors in vivo [32]. The P-DR5-B nanoparticles were found to surpass free DR5-B in cytotoxicity in all tested models. However, the advantage of P-DR5-B nanoparticles was more pronounced for the spheroids from DR5-*B*-resistant HT29 and MCF-7 cells (Figure 3b and Figure 4b, Table 2), while in the case of the spheroids from DR5-*B*-sensitive HCT116 cells the cytotoxicity of the P-DR5-B nanoparticles was comparable to that of free DR5-B (Figure 4a, Table 2). The cytotoxic effect was time-dependent, increasing from 24 to 48 h of incubation (Figure 3 and Figure 4, Table 2). Thus, conjugation of DR5-B to the nanoparticles resulted in an enhanced cell-killing capacity. The values of the standard error of the regression (S) for IC50 calculations are shown in Appendix A.

The morphology of 3D tumor spheroids of HCT116 colorectal carcinoma cells after incubation with P-DR5-B indicated apoptotic cell death (Appendix A).

The P-DR5-B nanoparticle’s safety and tumor specificity were determined by evaluating their cytotoxicity towards normal (non-tumor) cells, namely human keratinocytes HaCaT (Figure 5a) and human skin fibroblasts BJ-5ta (Figure 5b). Both DR5-B and blank Amph-PVP nanoparticles (without DR5-B) were non-toxic, while P-DR5-B nanoparticles were found to show a very small cytotoxic effect to BJ-5ta cells, which was beyond subtoxic values even at the maximal concentration (Figure 5a,b).

## 4. Discussion

The versatility and the unique properties of the PVP determined its wide use for various pharmaceutical and biomedical applications. Polymers based on *N*-vinylpyrrolidone derivatives are widely used in various medical devices and preparations, including the blood-retina nanocarrier platforms [33] and bioprinting [34]. PVP is used in drug delivery systems as a coating ingredient or in nanocarriers for the delivery of low-molecular-weight and hydrophobic drugs [8,35]. PVP is well-suited for bioconjugation as a drug carrier with a narrow molecular weight distribution [36]. Recently, it was proposed as a polymeric drug carrier for bioactive protein. For example, TNF-α conjugated with PVP circulated longer than TNF-α conjugated with polyethylene glycol with the same molecular weight [37]. PVP coating of the silver nanoparticles was used to reduce Ag cytotoxicity and to enhance stability in vivo, while further conjugation to IL-10 was found to increase the protein stability and anti-inflammatory efficacy [38].

Amph-PVP-based nanocarriers were proposed for the delivery of non-steroidal anti-inflammatory drugs [39,40] and for drug delivery into the cell nucleus [41]. Moreover, amphiphilic linear-dendritic block copolymers based on the linear PVP and dendritic phenylalanyl-lysine dipeptide were reported for the enzyme-responsive release of doxorubicin [42]. However, we did not find any reports, where Amph-PVP-based nanocarriers were proposed for targeted hydrophilic therapeutic macromolecules, namely targeted proteins.

At present, many efforts have been made to exploit DR5-dependent apoptosis induction for cancer treatment. TRAIL, a natural ligand for DR5, is a transmembrane protein with the N-terminal cytoplasmic domain and the C-terminal extracellular domain, which can be proteolytically cleaved from the cell surface and maintain in the body both in soluble and in membrane-associated forms. However, membrane-bound TRAIL exhibits increased activity [43]. Therefore, various nanoscale delivery systems were developed to enhance the delivery of TRAIL and other DR5-targeting molecules to cancer cells [20]. Among them, several TRAIL-based micellar systems were obtained. For example, self-assembled micellar nanoparticles from a biodegradable cationic copolymer P(MDS-*co*-CES) binding TRAIL through hydrogen bonds, hydrophobic interaction, and van der Waals forces [44]; PEG-PE (polyethylene glycol-phosphatidylethanolamine) micelles with incorporated DM-PIT-1 (*N*-{[(2-hydroxy-5-nitrophenyl)amino]carbonothioyl}-3,5-dimethylbenzamide) and TRAIL attached to the micelle surface via pNP groups [45]; self-assembled micelles from biodegradable amphiphilic copolymers, mPEG-PLA (monomethoxyl poly(ethylene glycol)–*b*-poly(dl-lactide)) and COOH-PEG-PLA, coupled with the TRAIL protein by a carbodiimide method [46]; paclitaxel loaded waterborne polyurethane nanomicelles conjugated with TRAIL through hydrogen bonds (PTX-PU-TRAIL) [47]; self-assembled amphiphilic peptide nanostructures targeting DR5 with TRAIL-mimetic DR5-targeting peptide, acting as a dual-targeting and therapeutic agent [48].

In the current study, we propose a new nanoscaled delivery system based on the Amph-PVP nanoparticles coated with the covalently attached antitumor DR5-specific cytokine DR5-B, reported by the authors earlier [21]. Despite Amph-PVPs hold a great promise for conjugation with therapeutic molecules, no attempts were made to use them as nanocarriers for TRAIL receptor agonists. Therefore, our data are absolutely new for this type of nanoscaled system.

Mostly all available TRAIL-based nanodelivery systems deal with the wild-type TRAIL, therefore they interact with all five TRAIL receptors. It was previously shown that TRAIL binding to a complicated death and decoy receptor system can lead to the activation of non-canonical signaling pathways, displaying anti-apoptotic effects [49]. Therefore, nanocarrier conjugation with the DR5-selective cytokine DR5-B allows for the avoidance of this obstacle with the advantage of overcoming the receptor-dependent resistance of tumor cells to TRAIL. Notably, specific binding of DR5-B protein to the DR5 receptor provides active tumor targeting of P-DR5-B nanoparticles.

To stabilize the Amph-PVP nanoparticles, model substance prothionamide was entrapped into the hydrophobic core. Prothionamide was chosen due to its high hydrophobicity, lack of cytotoxic activity at concentrations used and was approved for clinical application [50]. However, our further study will proceed in loading the P-DR5-B nanoparticles hydrophobic core with a number of chemotherapeutics, especially those, which showed synergistic antitumor activity with DR5-B.

To obtain the P-DR5-B nanoparticles, the DR5-B protein was modified by the cysteine amino acid residue at the N-end for further covalent conjugation with Amph-PVP. This construct has several advantages: (a) the covalent bond ensures efficient protein coupling by standard click chemistry reaction, providing construct stability at storage; (b) the correct sterical orientation of the protein molecules due to strict N-terminal conjugation of DR5-B/V114C which provides efficient DR5 receptor binding.

Due to steric hindrances along with protein conjugation to the nanoparticle surface, it was reasonable to obtain the Amph-PVP nanoparticles from a mixture of unmodified and maleimide-modified polymeric chains. To optimize the amount of the bound protein, we constructed the nanoparticles which differed in ratios of maleimide-modified to unmodified Amph-PVP polymers. Expectedly, the nanoparticles with a greater proportion of maleimide-modified Amph-PVPs coupled more DR5-B/V114C protein. However, the relation was non-linear, indicating that further increase in the maleimide-modified polymer content may result in an insignificant increase in protein sorption capacity. Therefore the further study was carried out with the nanoparticles containing 1:1 unmodified to modified Amph-PVPs.

Our earlier works evidence that at polymer concentrations near its CMC (about 10^−7^ M) the much smaller size of the micelles (20–30 nm) can be obtained [14]. However, this is valid for plain micelles, unloaded with drugs. For drugs entrapment or immobilization on the surface of the nanoparticles, the polymers are taken in concentrations several orders higher than their CMC (10^−6^–10^−4^ M), when more complex aggregates of spherical morphology and larger sizes are formed [39]. The introduction of the drug into the system also influences the size of obtained nanoparticles [13,40]. Therefore, the obtained nanoparticle size (about 220 nm) correlates with our previous findings.

Cytotoxicity of the P-DR5-B nanoparticles was tested in vitro using 3D multicellular tumor spheroids. The tumor spheroids were generated by a simple RGD-induced cell self-assembly technique which was recently developed by us [28]. As well known, tumor spheroids are considered as a more relevant model for anticancer drug discovery compared to monolayer cultures [51], particularly for studying the antitumor activity of nanomedicines [52,53]. Due to their 3D structure, the tumor spheroids can mimic heterogeneity and microenvironment of small avascular solid tumors in vivo, including specific gene expression, cell-to-cell and cell-to extracellular matrix interactions, growth kinetics, metabolic rates and resistance to chemotherapy [54]. As a result, this 3D in vitro model allows us to get more complete information on cytotoxic effects, being an intermediate stage between 2D in vitro model (monolayer culture) and testing in animals. Moreover, the use of the tumor spheroids could reduce the number of animal tests, meeting the ethical rule of the 3Rs (Reduce, Replace, Refine).

In our study, the P-DR5-B nanoparticles not only retained the biological activity of free DR5-B but were also found to show enhanced antitumor activity in 2D and 3D in vitro models. Primarily, conjugation of TRAIL and other DR5-specific molecules with nanoscaled carriers pursues a goal to enhance ligand stability and pharmacokinetic profile, as well as to increase nanocarrier accumulation in the tumor due to enhanced permeability and retention (EPR) effect. However, it is equally important that ligand immobilization on the nanoparticle surface increases its valence and promotes DR5 receptor clustering on the tumor cell surface. Notably, the antitumor activity of the P-DR5-B nanoparticles exceeded that of free DR5-B even in vitro. The possible reason is that conjugation of DR5-B with the surface of the Amph-PVP nanoparticles increased its local concentration, leading to more efficient apoptosis signaling due to enhanced DR5 clustering. Recent studies support this phenomenon, demonstrating that TRAIL immobilization on the nanoparticle surface can mimic the endogenous membrane-bound form and promote high-order TRAIL oligomerization. These results related to superior DR5 clustering, enhanced DISC recruitment and more efficient apoptosis induction [43,55,56,57,58,59]. Further in vivo experiments could provide more accurate data on the pharmacokinetic profile of the P-DR5-B nanoparticles and their antitumor efficacy.

## 5. Conclusions

In the current study, we fabricated and characterized in terms of size and morphology novel Amph-PVP nanoparticles covalently attached to DR5-specific antitumor cytokine DR5-B for targeted delivery to tumor cells. The cytotoxic effects of the obtained P-DR5-B nanoparticles were tested both in 2D and 3D in vitro models using three tumor cell lines and two normal cell cultures. This nanoscaled delivery system was shown to combine advantages of the Amph-PVP nanoparticles and DR5-specific tumor targeting, together with efficient antitumor activity due to DR5-induced apoptosis induction.

Due to Amph-PVPs hydrophobic core drug-loading capacity, along with the possibility of standard click chemistry application for surface modification by therapeutic hydrophilic proteins, we can suggest that this approach is promising for the development of a versatile targeted nanoscaled delivery system.

## Figures and Tables

**Figure 1 pharmaceutics-13-01413-f001:**
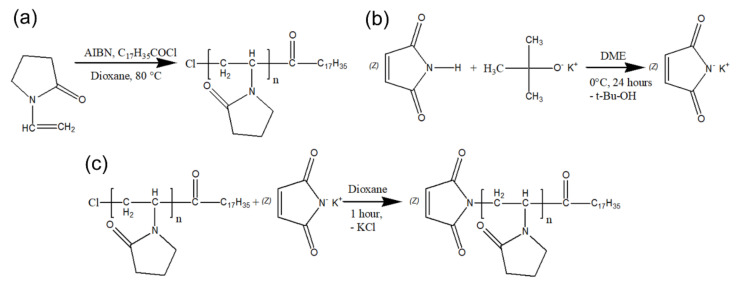
The synthesis schemes of compounds for further preparation of Amph-PVP nanoparticles: (**a**) Amph-PVP-Cl polymer; (**b**) potassium maleimide; (**c**) Amph-PVP-Cl modification by maleimide group.

**Figure 2 pharmaceutics-13-01413-f002:**
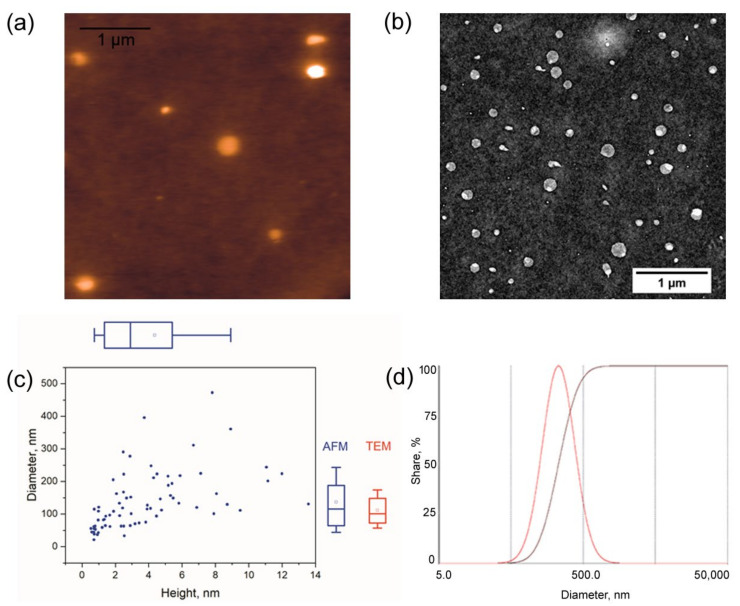
Characterization of DR5-*B*-conjugated Amph-PVP nanoparticles: (**a**) AFM image (Z range 15 nm); (**b**) TEM image; (**c**) diagram of the heights and widths of individual nanoparticles (blue, AFM data) and the diameters measured using TEM (in red); (**d**) hydrodynamic size distribution according to DLS. The box-and-whisker plots show the 10, 25, 50, 75 and 90 percentiles; the square mark shows the mean.

**Figure 3 pharmaceutics-13-01413-f003:**
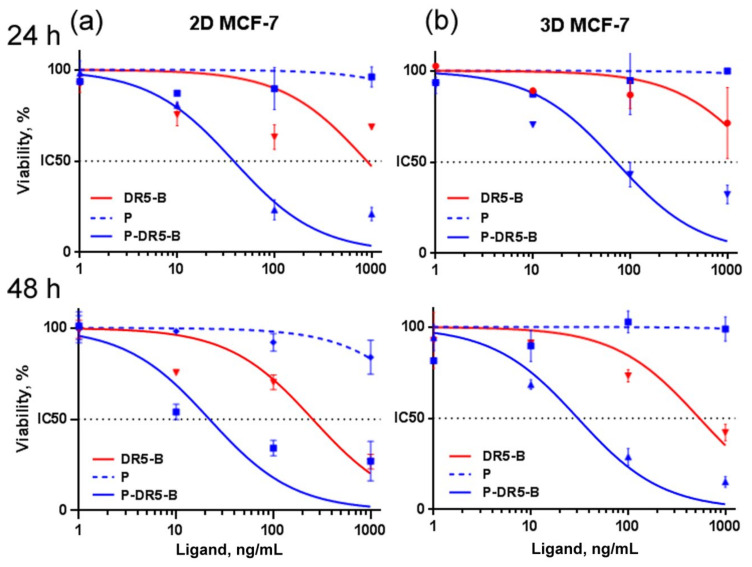
Cytotoxicity of the P-DR5-B nanoparticles in (**a**) 2D and (**b**) 3D human tumor models of breast adenocarcinoma MCF-7 in vitro for 24 and 48 h. Cell viability was determined by the MTT test. Mean ± Standard Deviation (*n* = 3).

**Figure 4 pharmaceutics-13-01413-f004:**
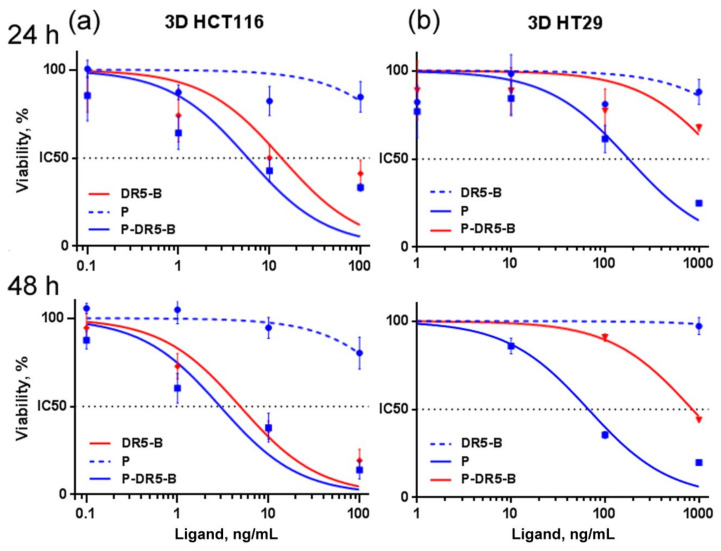
Cytotoxicity of the P-DR5-B nanoparticles in 3D human tumor models of (**a**) colorectal carcinoma HCT116; (**b**) colorectal adenocarcinoma HT29 in vitro for 24 and 48 h. Cell viability was determined by the MTT test. Mean ± Standard Deviation (*n* = 3).

**Figure 5 pharmaceutics-13-01413-f005:**
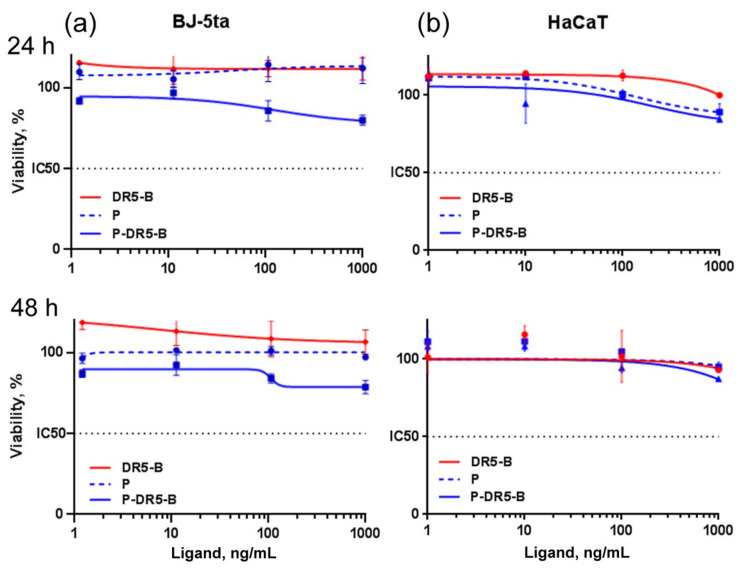
Cytotoxicity of the P-DR5-B nanoparticles in normal BJ-5ta (**a**) and HaCaT (**b**) cell lines after 24 and 48 h. Cell viability was determined by the MTT-test. Mean ± Standard Deviation (*n* = 3).

**Table 1 pharmaceutics-13-01413-t001:** Size, polydispersity and sorption capacity of Amph-PVP-based nanoparticles depending on the polymer composition.

Samples	P		P-DR5-B	
maleimide-modified to unmodified Amph-PVPs molar ratio	1:3	1:1	1:3	1:1
median size, nm ^1^	217.6	205.6	228.5	228.2
polydispersity index ^2^	0.299	0.302	0.274	0.303
amount of DR5-B, μg per 1 mg of the lyophilized nanoparticles	-	-	4.0 ± 0.06	5.0 ± 0.05

^1^ Size was measured by DLS; mean standard deviation (SD) of size ±8.7 nm. ^2^ Polydispersity index (PDI) was measured by DLS; mean SD of PDI ± 0.032.

**Table 2 pharmaceutics-13-01413-t002:** IC50 values (ng/mL) of free DR5-B and the P-DR5-B nanoparticles in 2D and 3D in vitro models.

Ligand	2D MCF-7		3D MCF-7		3D HCT116		3D HT29	
Incubation Time, h	24	48	24	48	24	48	24	48
DR5-B	901.3	275.5	>1000	584.0	16.3	5.0	>1000	863.1
P-DR5-B	39.8	21.6	72.1	31.7	5.9	2.8	184.2	68.4

## Data Availability

The data presented in this study are available on request from the corresponding author.

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
