# Peer review of "Amphiphilic Poly(N-vinylpyrrolidone) Nanoparticles Conjugated with DR5-Specific Antitumor Cytokine DR5-B for Targeted Delivery to Cancer Cells"

_pharmaceutics, 2021, doi:10.3390/pharmaceutics13091413_

Round 1

Reviewer 1 Report

In this contribution by Yagolovich and co-workers, the authors prepared amphiphilic poly(N-vinylyrrolidone) nanoparticles conjugated with DR5-specific antitumor cytokine DR5-B for targeted delivery to cancer cells. The results are kind of interesting and potentially attractive to the readership of Pharmaceutics. However, lo It could be publishable in due course but these points below must be addressed prior to publication.

  1. Line 25, the full name of TRAIL should be added as it appears first time.
  2. In the introduction, it will be better if the authors could discuss in a wider context, such as adding some discussion about using other targeted polymeric micelles drug delivery. Several studies (doi.org/10.1021/acs.biomac.6b00168; doi.org/10.1021/nn402662d; doi.org/10.1016/j.colsurfb.2020.111225) should be included.
  3. During the preparation of Amph-PVP-Cl, is there possible that some of the polymer is without stearoyl chloride?
  4. What is polydispersity of PVP (Mw/Mn)?
  5. The authors should add the 1H-NMR spectra of PVP-Cl and PVP-Cl modification by maleimide group.
  6. If the self-assemble condition is perfect, the size of micelles will be much smaller (ca. 10-20 nm) considering that the hydrophobic section is only C18 and the hydrophilic section is 10k of PVP. The authors must add discussion concerning to this point.
  7. For the cytotoxicity experiment, it is better that the author will include some typical images about the morphology of cells during the incubation with nanoparticles.
  8. Format issue. For example, ‘IC50’ (line 301).

Author Response

Thank you very much for thoroughly reviewing our manuscript. We have taken into account all the comments and have made the appropriate changes in the manuscript.

Point 1: Line 25, the full name of TRAIL should be added as it appears first time.

Response 1: We defined the full name of TRAIL in the abstract of the manuscript (Line 26).

Point 2: In the introduction, it will be better if the authors could discuss in a wider context, such as adding some discussion about using other targeted polymeric micelles drug delivery. Several studies (doi.org/10.1021/acs.biomac.6b00168; doi.org/10.1021/nn402662d; doi.org/10.1016/j.colsurfb.2020.111225) should be included.

Response 2: Thanks for your comment. We have added the appropriate discussion about usage of other targeted polymeric micelles for drug delivery to the introduction section. The following paragraph has been introduced in the text: “Presently, nanocarriers are increasingly developed…” (Line 43).

Point 3: During the preparation of Amph-PVP-Cl, is there possible that some of the polymer is without stearoyl chloride?

Response 3: First, we would like to mention that synthesis of amphiphilic N-vinylpyrrolidone polymers is studied by our group for several years. The results related to synthesis mechanisms, kinetics and peculiarities are thoroughly investigated and published in several articles and thesis. Here we present references for some of them, where preparation of the polymers is already described and discussed [1-5]. So, synthesis of the initial polymer was not a key point of discussion for this particular study, and we took ready polymers, prepared and purified in our laboratory, similar as some groups use ready lipids for liposome preparation for their further experiments.

As for the question on stearoyl chloride, the application of chain growth regulators for N-vinylpyrrolidone radical polymerization is based on their higher ability and rates to react with initiator to form initiating radicals. So, in this case, as it was shown in our previous works, most of the resulting polymers are formed with participation of chain growth regulator, namely stearoyl chloride. Also, resulting polymers are purified by dialysis and any molecules of simple PVP will be detached during purification.

  1. Kuskov A.N., Luss A.L., Gritskova I.A., Shtilman M.I., Motyakin M.V., Levina I.I., Nechaeva A.M., Sizova O.Y., Tsatsakis A.M., Mezhuev Y.O. Kinetics and mechanism of synthesis of carboxyl-containing N-vinyl-2-pyrrolidone telehelics for pharmacological use. // Polymers 2021, 13, 2569.
  2. Kuskov A.N., Kulikov P.P., Goryachaya A.V., Shtilman M.I., Tzatzarakis M.N., Tsatsakis A.M., Velonia K. Self-assembled amphiphilic poly-N-vinylpyrrolidone nanoparticles as carriers for hydrophobic drugs: stability aspects. // Journal of Applied Polymer Science. 2018. V.135. N. 1. P. 45637.
  3. Kulikov P.P., Goryachaya A.V., Luss A.L., Shtilman M.I., Kuskov A.N. Amphiphilic poly-N-vinyl-2-pyrrolidone: synthesis, properties, nanoparticles. // Polymer Science. Series D. 2017. V. 10. N.3. P. 263-268.
  4. Kuskov A.N., Kulikov P.P., Luss A.L.,  Goryachaya A.V.,  Shtilman M.I. Preparation of polymer nanoparticles by self-assembling of amphiphilic poly-N-vinylpyrrolidone derivatives in aqueous media, // Russian Journal of Applied Chemistry. 2016. V. 89(9). P. 1461-1468.
  5. Kuskov A.N., Shtilman M.I., Tsatsakis A.M., Torchilin V.P., Yamskov I.A. Synthesis of amphiphilic polymers of N-vinylpyrrolidone and acrylamide with different structure. Russian Journal of Applied Chemistry. 2005. V. 78(5). P. 806-810.

Point 4: What is polydispersity of PVP (Mw/Mn)?

Response 4: As we mentioned in the answer to the previous question, the typical amphiphilic PVP, prepared and purified in our laboratory, was used for this study. For polymer used (PVP-OD11000), polydispersity (Mw/Mn value) was 1.18 as determined by high-performance liquid chromatography.

Point 5: The authors should add the 1H-NMR spectra of PVP-Cl and PVP-Cl modification by maleimide group.

Response 5: To control Amph-PVP-Cl modification by maleimide group, we used functional and elemental analysis on Cl before and after polymer modification, which showed absence of unreacted groups. 1H-NMR spectra of amphiphilic polymers can be found in our previous papers (references are given above) as their study was carried out before and was not the particular aim of the current study. But it should be mentioned that the NMR method in case of polymers is not very useful for quantitative determination of polymer modification. Functional and elemental analysis are more reliable in this case.

Point 6: If the self-assemble condition is perfect, the size of micelles will be much smaller (ca. 10-20 nm) considering that the hydrophobic section is only C18 and the hydrophilic section is 10k of PVP. The authors must add discussion concerning to this point.

Response 6: The optimal size of the micelles (20 nm) can be obtained at polymer concentrations near its CMC (about 10-7 M). We obtained such nanoparticles in our previous works for plain micelles, unloaded with drugs [1]. For drugs entrapment or immobilization on the surface of the nanoparticles, we take polymer at concentrations several orders higher than their CMC (10-6 -10-4 M), when more complex aggregates of spherical morphology and larger sizes are formed [2]. The introduction of the drug into the system also influences the size of obtained nanoparticles [3,4].

The influence of polymers structure (hydrophobic/hydrophilic blocks), concentration, pH, ionic force, temperature, drug type and ratio to polymer and many other factors on their behavior in aqueous media and characteristics of the resulted nanoparticles was also investigated and published previously. That is why we use term “nanoparticle” or “aggregate” rather than “micelles” to describe our polymeric carriers.

It should be also noted, that optimal size of nanocarriers for drug delivery to cells is considered to be 50-300 nm, meaning that DR5-B-conjugated polymeric nanoparticles obtained in our work fall into eligible size range.

The appropriate discussion on the size of the obtained DR5-B-conjugated polymeric nanoparticles has been added to the manuscript. Please see the paragraph “Our earlier works evidence…” in the discussion section (Line 419).

  1. Kuskov, A.N.; Shtilman, M.I.; Goryachaya, A.V.; Tashmuhamedov, R.I.; Yaroslavov, A.A.; Torchilin, V.P.; Tsatsakis, A.M.; Rizos, A.K. Self-Assembling Nanoscaled Drug Delivery Systems Composed of Amphiphilic Poly-N-Vinylpyrrolidones. Journal of Non-Crystalline Solids 2007, 353, 3969–3975, doi:10.1016/j.jnoncrysol.2007.02.061.
  2. Kuskov AN, Voskresenskaya AA, Goryachaya AV, Artyukhov AA, Shtilman MI, Tsatsakis AM. Preparation and characterization of amphiphilic poly-N-vinylpyrrolidone nanoparticles containing indomethacin. J Mater Sci Mater Med. 2010 May;21(5):1521-30. doi: 10.1007/s10856-010-4029-1.
  3. Kuskov, A.N.; Villemson, A.L.; Shtilman, M.I.; Larionova, N.I.; Tsatsakis, A.M.; Tsikalas, I.; Rizos, A.K. Amphiphilic Poly- N -Vinylpyrrolidone Nanocarriers with Incorporated Model Proteins. J. Phys.: Condens. Matter 2007, 19, 205139, doi:10.1088/0953-8984/19/20/205139.
  4. Kuskov, A.N.; Voskresenskaya, A.A.; Goryachaya, A.V.; Shtilman, M.I.; Spandidos, D.A.; Rizos, A.K.; Tsatsakis, A.M. Amphiphilic Poly-N-Vinylpyrrolidone Nanoparticles as Carriers for Non-Steroidal Anti-Inflammatory Drugs: Characterization and in Vitro Controlled Release of Indomethacin. Int J Mol Med 2010, 26, 85–94, doi:10.3892/ijmm_00000438.

Point 7: For the cytotoxicity experiment, it is better that the author will include some typical images about the morphology of cells during the incubation with nanoparticles.

Response 7: Thanks for your comment. The image of tumor cells undergoing apoptosis after the incubation with P-DR5-B nanoparticles has been added to the Supplementary materials (Figure S3). The appropriate sentence has been introduced in the results section of the manuscript: “The morphology of 3D tumor spheroids…” (Line 320).

Point 8: Format issue. For example, ‘IC50’ (line 301).

Response 8: If we understood this comment correctly, it is related to the imperfect formatting of the Table 2. To address this issue, the Table 2 has been accurately formatted in the manuscript.

Reviewer 2 Report

The authors prepared an amphiphilic PVP nanoparticles and conjugated them with DR5-B for anti-tumor effect. Overall, the manuscript is well-written. I have some comments that need to be addressed before the acceptance of the manuscript. 

1. At what concentration did the author measure the DLS results of nanoparticles?

2. The particle size of prepared amph-PVP-based nanoparticles is around 200 nm with a PdI around 0.3. Though author demonstrated that samples were unimodal on line 261, the PdI of 0.3 is not widely accepted in the field as unimodal. With this size and PdI, there are significant concerns on the structure of nanoparticle and sterility of nanoparticles. Is there any approach to decrease the PdI? How did the author sterilize the nanoparticles before dosing to the in vitro assays?

In addition, the author showed the non-linearship between DR5-B/V114C protein and ratio of maleimide-modified polymer, it raised further concern on the structure of nanoparticles. The observations indicate that some maleimide groups are shielded by other polymer chains, thereby, the amount of DR5-B conjugated on the nanoparticles is uncontrollable. However, amount of the DR5-B has a decisive impact on the therapeutic efficacy. My question is what the approach the authors will take to minimize this potential cause of batch difference? 

3. What's the serum stability of amph-PVP-based nanoparticles? There is a concern that these nanoparticles may be cleared fast after being administered into animals. I couldn't understand where the hindrance effect is coming from as indicated on line 402. Is there any evidence that protein conjugation can provide hindrance effect? 

4. I only see 4 concentration points in Figure 3, 4 and 5. The curves are clearly not well fitted with the data points. Can author show the R-square value of the fitting? With these very limited amount of data points, it's hard to predict the IC50 accurately. 

5. The 24-hour and 48-hour incubation used in cytotoxicity evaluation are too long, supposing that the nanoparticles will accumulate in tumor area after in vivo administration. What's planned in vivo administration route for these nanoparticles, as indicated on line 440?

6. In discussion, paragraphs 2-5 are repetitive to the introduction section. These paragraphs can be shrieked into one paragraph. It will be great if the authors can compare the system developed in this manuscript with other DR5/TRAIL-based micellar systems in the field. 

7. Last but not the least, in the title, it says "targeted delivery to cancer cells". However, I didn't see enough evidence for this targeted delivery. Can author justify this "targeted" please? 

Author Response

Thank you very much for careful reading our manuscript. We have taken into account all your comments and have made the appropriate corrections in the manuscript.

Point 1: At what concentration did the author measure the DLS results of nanoparticles?

Response 1: The DLS measurements were made at polymer concentration of 2 mg/ml, corresponding to the protein concentration of 10 µg/ml.

Point 2: The particle size of prepared amph-PVP-based nanoparticles is around 200 nm with a PdI around 0.3. Though author demonstrated that samples were unimodal on line 261, the PdI of 0.3 is not widely accepted in the field as unimodal. With this size and PdI, there are significant concerns on the structure of nanoparticle and sterility of nanoparticles. Is there any approach to decrease the PdI? How did the author sterilize the nanoparticles before dosing to the in vitro assays?

Response 2: In drug delivery applications using polymeric and lipid-based carriers, a PDI of 0.2-0.3 is considered to be acceptable [Danaei, M.; Dehghankhold, M.; Ataei, S.; Hasanzadeh Davarani, F.; Javanmard, R.; Dokhani, A.; Khorasani, S.; Mozafari, M.R. Impact of Particle Size and Polydispersity Index on the Clinical Applications of Lipidic Nanocarrier Systems. Pharmaceutics 2018, 10, 57. https://doi.org/10.3390/pharmaceutics10020057]. Also, PDI value of 0.3 is commonly stated by the manufacturers of the equipment for DLS measurement as the upper limit for unimodal distribution, so we took this value for the estimation.

The PDI and particle size may be optimized by changing the preparation conditions (ultrasonic homogenization intensity, length, polymer molecular weight, the ratio of the components and etc.).

So, it is a large amount of work to do and we, no doubt, are planning to optimize the conditions which will be reflected in our further research. Also, we will try to decrease the size of the prepared nanoparticles.

As for sterility concern, the nanoparticles were routinely sterilized by autoclaving before conjugation with DR5-B protein. Autoclaving does not significantly affect particle size and distribution. All subsequent procedures were carried out under sterile conditions.

In addition, the author showed the non-linearship between DR5-B/V114C protein and ratio of maleimide-modified polymer, it raised further concern on the structure of nanoparticles. The observations indicate that some maleimide groups are shielded by other polymer chains, thereby, the amount of DR5-B conjugated on the nanoparticles is uncontrollable. However, amount of the DR5-B has a decisive impact on the therapeutic efficacy. My question is what the approach the authors will take to minimize this potential cause of batch difference? 

Response: In our opinion, the aforementioned non-linearship is related not to shielding of maleimide groups by other polymer chains, but rather to shielding by large protein molecules, comprising about 58 kDa for DR5-B trimer. This could lead to steric hindrances upon protein binding to the nanoparticle surface. That is, even if all polymer molecules will be maleimide-modified (instead of 1:1 maleimide-modified to unmodified Amph-PVPs molar ratio), this won’t result in double increase of sorption capacity.

For this reason, the quantity of protein molecules that can be sterically fitted at the nanoparticle surface is a quite constant value. This excludes the variability in nanoparticle sorption capacity and, as a result, the risk of batch difference upon protein conjugation.

In our further studies, to increase the availability of maleimide groups, we are planning to use two different polymers, a not-modified polymer with low molecular weight as a basis for micelles formation and a modified polymer with a high molecular weight (12 kDa and higher) for protein linkage.

Point 3: What's the serum stability of amph-PVP-based nanoparticles? There is a concern that these nanoparticles may be cleared fast after being administered into animals. I couldn't understand where the hindrance effect is coming from as indicated on line 402. Is there any evidence that protein conjugation can provide hindrance effect? 

Response 3: The serum stability of Amph-PVP nanoparticles  has been reported previously [1]. The appropriate sentence is added to the introduction section (please see “The Amph-PVP nanoparticles show serum stability…”, Line 54).

On the contrary, as well known, the conjugation of TRAIL-based proteins to the nanocarriers increases serum stability and lowers the renal clearance of the protein [2-4].

The hindrance effect indicated on line 402 of the initial manuscript is related to steric effects upon protein binding to the micelle surface, as it has been mentioned above.

  1. Tsatsakis A, Stratidakis AK, Goryachaya AV, Tzatzarakis MN, Stivaktakis PD, Docea AO, Berdiaki A, Nikitovic D, Velonia K, Shtilman MI, Rizos AK, Kuskov AN. In vitro blood compatibility and in vitro cytotoxicity of amphiphilic poly-N-vinylpyrrolidone nanoparticles. Food Chem Toxicol. 2019 May;127:42-52. doi: 10.1016/j.fct.2019.02.041.
  2. Loi M, Becherini P, Emionite L, Giacomini A, Cossu I, Destefanis E, Brignole C, Di Paolo D, Piaggio F, Perri P, Cilli M, Pastorino F, Ponzoni M. sTRAIL coupled to liposomes improves its pharmacokinetic profile and overcomes neuroblastoma tumour resistance in combination with Bortezomib. J Control Release. 2014 Oct 28;192:157-66. doi: 10.1016/j.jconrel.2014.07.009. Epub 2014 Jul 17.
  3. Kim TH, Jiang HH, Youn YS, Park CW, Lim SM, Jin CH, Tak KK, Lee HS, Lee KC. Preparation and characterization of Apo2L/TNF-related apoptosis-inducing ligand-loaded human serum albumin nanoparticles with improved stability and tumor distribution. J Pharm Sci. 2011 Feb;100(2):482-91. doi: 10.1002/jps.22298.
  4. Kim TH, Jiang HH, Park CW, Youn YS, Lee S, Chen X, Lee KC. PEGylated TNF-related apoptosis-inducing ligand (TRAIL)-loaded sustained release PLGA microspheres for enhanced stability and antitumor activity. J Control Release. 2011 Feb 28;150(1):63-9. doi: 10.1016/j.jconrel.2010.10.037.

Point 4: I only see 4 concentration points in Figure 3, 4 and 5. The curves are clearly not well fitted with the data points. Can author show the R-square value of the fitting? With these very limited amount of data points, it's hard to predict the IC50 accurately. 

Response 4: The cytotoxicity assay was carried out using MTT test with at least three independent repeats. The experimental results were transferred to GraphPad Prism Software, which contains a special algorithm for processing cytotoxicity data (namely «Dose-response – Inhibition: log(inhibitor) vs. response – Variable sloop (four parameters)» algorithm). This approach is widely used and accepted, since it allows to perform a more accurate data analysis, compared to direct curve plotting using separate points (for example using Microsoft Office Excel Software).  GraphPad software allows us to compare our obtained results with those of other researchers, since they present the MTT data in a similar way. This approach has also been used in our previously published papers [1, 2].

  1. Akasov R, Borodina T, Zaytseva E, Sumina A, Bukreeva T, Burov S, Markvicheva E. Ultrasonically Assisted Polysaccharide Microcontainers for Delivery of Lipophilic Antitumor Drugs: Preparation and in Vitro Evaluation. ACS Appl Mater Interfaces. 2015 Aug 5;7(30):16581-9. doi: 10.1021/acsami.5b04141.
  2. Gileva A, Sarychev G, Kondrya U, Mironova M, Sapach A, Selina O, Budanova U, Burov S, Sebyakin Y, Markvicheva E. Lipoamino acid-based cerasomes for doxorubicin delivery: Preparation and in vitro evaluation. Mater Sci Eng C Mater Biol Appl. 2019 Jul;100:724-734. doi: 10.1016/j.msec.2019.02.111.

Point 5: The 24-hour and 48-hour incubation used in cytotoxicity evaluation are too long, supposing that the nanoparticles will accumulate in tumor area after in vivo administration. What's planned in vivo administration route for these nanoparticles, as indicated on line 440?

Response 5: Incubation for 24-, 48- and 72-hour is a standard approach to study cytotoxicity of any nanocarriers. The cytotoxicity of free TRAIL protein is commonly evaluated at least after 24 hours. As for in vivo administration route, our previous studies of Amph-PVP serum stability let us expect that intravenous injection of nanoparticles with DR5-B could be well tolerated and rather effective. However, we also can consider intratumoral injection as an alternative administration route.

Point 6: In discussion, paragraphs 2-5 are repetitive to the introduction section. These paragraphs can be shrieked into one paragraph. It will be great if the authors can compare the system developed in this manuscript with other DR5/TRAIL-based micellar systems in the field. 

Response 6: Thank you, we agree with this comment. We have reduced the discussion section, and have added several sentences about other DR5/TRAIL-based micellar systems. Please see “Among them, several TRAIL-based micellar systems …” (Line 371).

Point 7: Last but not the least, in the title, it says "targeted delivery to cancer cells". However, I didn't see enough evidence for this targeted delivery. Can author justify this "targeted" please? 

Response 7: The active targeting of DR5-B-conjugated nanoparticles is provided by specific binding of DR5-B protein to DR5 receptor, which is expressed mainly by tumor cells. The appropriate sentence has been introduced in the text: “Notably, specific binding of DR5-B protein to the DR5 receptor provides active tumor targeting of P-DR5-B nanoparticles” (Line 395)

As for normal cells, we have demonstrated that DR5-B-conjugated nanoparticles are non-toxic towards non-tumor human keratinocytes HaCaT and human skin fibroblasts BJ-5ta.

Round 2

Reviewer 1 Report

The ms. is ready for published in Pharmaceutics.

Author Response

We are grateful for attention to our manuscript. 

Reviewer 2 Report

I would like to thank the authors for the responses. I think most of the points have been addressed. However, I still would like to follow-up on point 4 on the R-square value of the fitting. You can fit any series of data points using graphpad. But that doesn't mean the fitting is reasonable or the prediction is accurate. You have to show the R-square to prove the model you used in suitable. 

Author Response

Point: I would like to thank the authors for the responses. I think most of the points have been addressed. However, I still would like to follow-up on point 4 on the R-square value of the fitting. You can fit any series of data points using graphpad. But that doesn't mean the fitting is reasonable or the prediction is accurate. You have to show the R-square to prove the model you used in suitable. 

Response: We are grateful for thorough attention to our manuscript. We agree that additional information is needed to prove the suitability of the model which we used.

In our work, we used non-linear regression analysis to determine IC50, which is accepted as most accurate and practicable to fit biochemical data [1]. However, earlier it was shown that R-squared doesn’t work out correctly for non-linear regression models [2] (despite this, most statistical software still calculate R-squared for nonlinear models even though it is statistically incorrect). The other goodness-of-fit measures are better suited for non-linear regression model we used, for example standard error of the regression (S). Also, the S value is more useful to know than the R-squared of the model because it provides us with actual units of the response variable. 

We have added the Table S1 with the corresponding S values to the Supplementary file and pointed this out in the manuscript (line 319).

  1. Leatherbarrow RJ. Using linear and non-linear regression to fit biochemical data. Trends Biochem Sci. 1990 Dec;15(12):455-8. doi: 10.1016/0968-0004(90)90295-m. PMID: 2077683.
  2. Spiess AN, Neumeyer N. An evaluation of R2 as an inadequate measure for nonlinear models in pharmacological and biochemical research: a Monte Carlo approach. BMC Pharmacol. 2010 Jun 7;10:6. doi: 10.1186/1471-2210-10-6
